# Some CSF Kynurenine Pathway Intermediates Associated with Disease Evolution in Amyotrophic Lateral Sclerosis

**DOI:** 10.3390/biom11050691

**Published:** 2021-05-05

**Authors:** Hugo Alarcan, Romane Chaumond, Patrick Emond, Isabelle Benz-De Bretagne, Antoine Lefèvre, Salah-eddine Bakkouche, Charlotte Veyrat-Durebex, Patrick Vourc’h, Christian Andres, Philippe Corcia, Hélène Blasco

**Affiliations:** 1Laboratoire de Biochimie et Biologie Moléculaire, CHRU Bretonneau, 2 Boulevard Tonnellé, 37000 Tours, France; hugo.alarcan@etu.univ-tours.fr (H.A.); debucquet.romane@gmail.com (R.C.); corcia@med.univ-tours.fr (I.B.-D.B.); C.VEYRATDUREBEX@chu-tours.fr (C.V.-D.); patrick.vourch@univ-tours.fr (P.V.); christian.andres@univ-tours.fr (C.A.); 2UMR 1253 iBrain, Université de Tours, Inserm, 10 Boulevard Tonnellé, 37000 Tours, France; emond@univ-tours.fr (P.E.); antoine.lefevre49@gmail.com (A.L.); philippe.corcia@univ-tours.fr (P.C.); 3Laboratoire de Médecine Nucléaire In Vitro, CHRU Bretonneau, 2 Boulevard Tonnellé, 37000 Tours, France; 4Service de Neurologie, CHRU Bretonneau, 2 Boulevard Tonnellé, 37000 Tours, France; S.BAKKOUCHE@chu-tours.fr

**Keywords:** amyotrophic lateral sclerosis, kynurenine pathway, tryptophan, amino acids, cerebrospinal fluid, PLS-DA

## Abstract

The aim of this study was to evaluate the kynurenine pathway (KP) and amino acids profile, using mass spectrometry, in the cerebrospinal fluid (CSF) of 42 amyotrophic lateral sclerosis (ALS) patients at the diagnosis and 40 controls to detect early disorders of these pathways. Diagnostic and predictive ability (based on weight loss, forced vital capacity, ALS Functional Rating Scale—Revised evolution over 12 months, and survival time) of these metabolites were evaluated using univariate followed by supervised multivariate analysis. The multivariate model between ALS and controls was not significant but highlighted some KP metabolites (kynurenine (KYN), kynurenic acid (KYNA), 3-Hydroxynurenine (3-HK)/KYNA ratio), and amino acids (Lysine, asparagine) as involved in the discrimination between groups (accuracy 62%). It revealed a probable KP impairment toward neurotoxicity in ALS patients and in bulbar forms. Regarding the prognostic effect of metabolites, 12 were commonly discriminant for at least 3 of 4 disease evolution criteria. This investigation was crucial as it did not show significant changes in CSF concentrations of amino acids and KP intermediates in early ALS evolution. However, trends of KP modifications suggest further exploration. The unclear kinetics of neuroinflammation linked to KP support the interest in exploring these pathways during disease evolution through a longitudinal strategy.

## 1. Introduction

Amyotrophic lateral sclerosis (ALS), the most common form of motor neuron disorder in adults, is characterized by degeneration of both upper and lower motor neurons in bulbar and spinal territories. Initial presentation of ALS may vary between patients, most presenting a spinal-onset disease, though bulbar or respiratory onset can also be observed. As no effective treatment is available, disease evolution is rapid, and death typically occurs after 3 to 5 years of evolution due to respiratory paralysis [1].

To date, diagnosis of ALS relies on the Airlie House criteria [2]. Diagnosis according to these criteria requires elimination of other neurological diseases and relies primarily on clinical criteria and progression of motor weakness. To date there is no diagnostic or prognostic biomarker available in clinical practice. Disease heterogeneity and complex pathogenesis hamper the findings of both reproducible biomarkers and therapies. Thus, there is an urgent need to apply new biomarker research strategies to find relevant biological compounds helpful for both diagnostic and prognostic concerns, but also to achieve a better understanding of the disease pathogenesis. The pathophysiological mechanisms underlying ALS remain partially understood. Several mechanisms contribute to the initiation and progression of ALS, including, among others, aggregation and accumulation of ubiquitylated proteins inclusions in motoneurons, alterations of mRNA processing, glutamate-mediated excitotoxicity, oxidative stress, mitochondrial dysfunction, and neuroinflammation [2]. Importantly, numerous studies have reported metabolic alterations, including hypermetabolism and energetic substrate metabolism alterations [3].

Among the informative biological avenues recently investigated in neuromuscular disorders (NMD), the kynurenine (KYN) Pathway (KP) appears as one of the most promising [4,5,6]. KP represents the major route of tryptophan (TRP) catabolism to nicotinamide adenine dinucleotide (NAD+) [7]. Briefly, in the brain, TRP catabolism is initiated by indoleamine 2,3 dioxygenase 1 (IDO-1), some of whose intermediates could have either neuroprotective or neurotoxic roles. Neuroprotective intermediates are kynurenic acid (KYNA), a N-methyl-D-aspartic (NDMA) receptor antagonist and picolinic acid (PIC), while 3-hydroxykynurenine (3-HK) and quinolinic acid (QUIN), a NMDA receptor agonist, are neurotoxic [5]. In the context of inflammation, larger amounts of NAD+ are needed by immune cells. To meet this requirement, IDO-1 expression is upregulated by proinflammatory cytokines, mainly interferon gamma (IFN-γ), interferon alpha (IFN-α), and interleukin 6 (IL-6) [7,8,9]. Activation of IDO-1 and production of KP intermediates in turn have immunomodulatory effects [8,9]. Contrasting with TRP and kynurenine which cross the blood–brain barrier and can be elevated in cerebrospinal fluid (CSF) after systemic inflammation, KYNA and QUIN cross this barrier poorly, are produced locally in situ from kynurenine, and their elevation reflects neuroinflammation [7]. Thus, this pathway represents a bridge between inflammation and metabolism alterations, in consistence with the predominance of these mechanisms in ALS. Importantly, impaired KP has previously been reported in patients suffering from multiple sclerosis, Huntington’s, Parkinson’ disease (PD), and Alzheimer’s disease (AD) [6]. To our knowledge, very few reports have investigated KP in ALS patients and those that have presented controversial results [10,11,12,13].

In this context, the aim of this study was to measure amino acids and intermediates of tryptophan metabolism (including KP) in the CSF of ALS patients at the time of diagnosis to identify new early diagnostic or prognostic biomarkers. This work also provided an opportunity to deeply investigate the metabolism focused on this promising pathway, via the new angle of targeted metabolomics.

## 2. Materials and Methods

### 2.1. Patients and Controls

CSF samples were obtained from ALS patients and from gender- and age-matched controls at the time of diagnosis and then stored at −80 °C. Analytical methods were carried out in accordance with relevant guidelines. All experimental protocols were approved by the University Hospital of Tours. All patients included in this study were informed in writing regarding the collection of their samples remaining from routine biological analyses for research aims and given the right to refuse such uses; however, none refused. Moreover, all patients were informed about the data obtained and about their right to access these data, according to articles L.1121-1 and R1121-2 of the French Public Health Code (CNIL n°2019-071).

All ALS patients met the revised criteria to establish the diagnosis [14]. The control group included subjects with other neurologic diseases, mostly neuropathy without etiology or iatrogenic (n = 36), chronic inflammatory demyelinating polyradiculoneuropathies (n = 3), or neuropathies associated with dysglobulinemia (n = 3). Information concerning diagnosis, sex, age, site at onset (spinal, bulbar, respiratory), and date of initial symptoms were obtained for each patient. For estimation of disease progression, Body Mass Index (BMI), ALS Functional Rating Scale—Revised (ASLFRS-r) score and Forced Vital Capacity (FVC) were documented at the time of diagnosis and at each medical visit over 12 months. The date of death was also collected when available and duration of ALS was defined as the time between the appearance of the first symptoms of the disease and death. Disease progression was analysed according to the percentage of BMI, ALSFRS-r and FVC modification over 12 months (and expressed as classes, according to the median value of the parameter) and the survival duration.

### 2.2. CSF Amino Acid Quantification

Sample preparation was performed as previously presented and approved by the French Accreditation Committee (Cofrac, NF EN ISO 15189). First, 100 µL of precipitation reagent were added to 50 µL of the patient’s CSF. The mix was then vortexed for one minute to optimize protein precipitation and centrifuged for 10 min at 3000 *g* at 5 ± 3 °C. Ten microliters of supernatant were transferred to an empty mass spectrometry glass vial. Seventy microliters of borate buffer and 20 µL of derivatization reagent were added and the sample was incubated for 10 min at 55 °C.

Samples were analyzed on a Waters^®^ (Millford, MA, USA) ultra-high-performance liquid chromatography (UHPLC) system consisting of a UHPLC quaternary solvent manager (QSM), a flow-through-needle manager injector (FTN), a column manager oven (CM) and a single quadrupole mass detector (QDa). Separation was performed on a Cortecs ultra-performance liquid chromatography (UPLC) C18 (1.6 µm, 2.1 × 150 mm) column maintained at 55 °C during the analysis. The volume injected onto the column was 2 µL. Gradient elution was performed at constant flow rate (0.5 mL/min) using 0.1% formic acid in water as mobile phase A and 0.1% formic acid in acetonitrile as mobile phase B. Gradient started at 99% A and 1% B for 1 min. Mobile phase B was then increased linearly (curve 6) to 13% in 1 min. Then mobile phase B was increased to 15% in 4.5 min and finally to 95% in 2 min. This condition was held for 1 min then returned to initial condition over 2.5 min.

### 2.3. Kynurenine Pathway Exploration

For the analysis of tryptophan metabolites, 100 µL of CSF were added to 100 µL of a solution of internal standards and 300 µL of methanol. To build the calibration curves, standards followed the same pre-analytical treatment. After stirring and incubation for 30 min at −20 °C, each sample was centrifuged (15,000 g for 15 min at 4 °C). The supernatant (300 μL) was then transferred to a 96-well plate. After simultaneous evaporation, each well was resuspended in 100 μL of a methanol/water mixture (10/90). Finally, 5 μL were injected into the LC-MS (XEVO-TQ-XS, Waters^®^). A Kinetex C18 xb column (1.7 μm × 150 mm × 2.1 mm, temperature 55 °C) associated with a gradient of two mobile phases (Phase A: Water + 0.5% formic acid; Phase B: MeOH + 0.5% formic acid) at a flow rate of 0.4 mL/min was used. Some chromatograms and calibration curves, the solvent gradient, and the parameters of the Unispray^®^ ion source, along with the parameters for fragmentation, and analysis of the metabolites and internal standards, are provided in the Appendix A (Appendix A).

For each metabolite, a calibration curve was created by calculating the intensity ratio obtained between the metabolite and its internal standard. These calibration curves were then used to determine the concentrations of each metabolite in patient samples.

IDO-1 enzyme activity was estimated by the KYN/TRP ratio calculation as described elsewhere [10]. The use of metabolite ratios (KYNA/KYN, 3-HK/KYN, QUIN/KYNA, QUIN/KYN, 3-HK/KYNA) allowed us to not rely only on absolute levels and to determine reflected activities of other enzymes, as well as the relationship between neurotoxic and neuroprotective compounds [6,15].

### 2.4. Statistical Analysis

The study had three objectives: To find new biomarkers (1) for the diagnosis of ALS by comparison of ALS patients with controls, (2) to better characterize the disease, such as the site of onset, age of first symptoms, ASLFRS-r, FVC, and weight loss, and (3) to improve prognosis by comparing subsets of ALS patients. We used a two-step approach for each analytical strategy based on univariate analysis followed by multivariate analysis to consider a profile of biomarkers and not each metabolite independent from the others.

#### 2.4.1. Univariate Analysis

Univariates analyses of metabolite concentrations between two groups were performed using the non-parametric Wilcoxon test. *p*-values were corrected for multiple statistical tests using the false discovery rate (FDR) method. Volcano plots were constructed using MetaboAnalyst. 5.0 (www.metaboanalyst.ca/faces/home.xhtml). Metabolites with a fold change (FC) threshold of 1.2 and a raw *p*-value threshold of 0.2 were highlighted. FCs between the two groups are represented by the x-axis (log scale) and the raw *p*-value for the Wilcoxon -test is represented by the y-axis (negative log scale).

#### 2.4.2. Multivariate Analysis

MetaboAnalyst 5.0 was also used for multivariate analysis. After normalization of the variables by summing the peaks, log transformation, and autoscaling, unsupervised multivariate analysis based on principal component analyses (PCA) was performed. Grouping patterns, trends, and outliers were examined on scatter plots. Then, partial least squares discriminant analyses (PLS-DA) were conducted. PLS-DA models were cross-validated by withholding 1/10 of the samples in ten simulations (each sample being omitted once) to avoid overfitting. To test the relevance of these selected compounds, the quality of the model built from them was assessed by prediction accuracy and permutation tests. The performance measures of the permutated data usually form a normal distribution, and if the performance score of the original data lies outside the distribution, then the results are considered to be significant. A variable influence on projection (VIP) value was given to each variable. Features with A VIP > 0.8 were considered important to the model. Venn diagrams were then drawn using Venny 2.1 (https://bioinfogp.cnb.csic.es/tools/venny/), to highlight the metabolites most significantly associated with the diagnosis or disease evolution.

Multiparametric survival analysis of disease duration was performed on metabolites commonly important (according to their VIP value) to at least 3 out of the 4 evolution criteria. Univariate analysis was first conducted on each metabolite using the Cox model. Those with a *p*-value < 0.2 were included in the multivariate test, after elimination of features highly correlated between them. JMP statistical software version 13.0 (SAS Institute, Cary, NC) was used for this analysis.

## 3. Results

### 3.1. Patient Characteristics

Eighty-two patients were included in this study, comprising 40 ALS patients and 42 control subjects matched by age and sex. Clinical characteristics of the patients are reported in Table 1. There was no difference for gender and age between the two groups. Mean age at onset for ALS patients was 64.4 ±12.5 years and 42.5% were females. The site of onset was spinal in 72.5% of patients and bulbar in 27.5%. No respiratory onset was recorded. The median duration of the disease was 25.0 months (18.1–38.8). Values of ASLFRS-r, weight, and FVC at 12 months were missing for 40%, 42.5%, and 72.5% of patients, respectively.

As expected, the duration of disease correlated with age of onset (*p* < 0.002), weight loss (*p* < 0.001), and FVC at diagnosis (0.03). Using multivariate analysis, only age of onset (*p* = 0.04) and weight loss (*p* = 0.04) remained meaningful.

### 3.2. Amino Acids and Metabolites of the KP

CSF concentrations of 22 amino acids were compared between ALS and controls. Cystine and homocysteine, available with the method, were below the limit of quantification (LOQ) for all subjects. Mean concentrations with standard deviation of each compound are represented in Table 2. Mean values are in the range of normal concentration for both control and ALS patients, with the exception of glutamic acid (normal concentration <4 µM).

The profile of tryptophan pathway intermediates was not available for 4 ALS and 1 control subject due to an insufficient remaining volume of CSF. Of the 20 metabolites measurable by LC-MS/MS, 11 were detected and quantified in our patients, including 3-HK, QUIN, serotonin, 5-OH tryptophan, KYN, TRP, 5-OH indole acetic acid (5-HIAA), KYNA, indole-3-lactic acid, indole-3-aldehyde, and indole-3-acetic acid. Of note, PIC, anthranilic acid, and xanthurenic acid were below the limit of quantification for all our subjects. Mean concentrations with standard deviation for each compound are represented in Table 2. Figure 1 represents the metabolic pathway primarily explored in this study.

### 3.3. No relevant Diagnosis Biomarker among Amino Acids or Metabolites from KP

Using univariate analysis, no compound, neither amino acids nor tryptophan pathway intermediates, differed significantly between ALS and control subjects after FDR correction of *p*-values. The volcano plot revealed the following compounds as the most relevant: KYNA (FC: 0.64, *p* = 0.029), KYN (FC: 0.78, *p* = 0.036), and QUIN (FC: 0.78, *p* = 0.044). These compounds were lower in ALS subjects, while 3-HK/KYNA (FC: 1.68, *p* = 0.067) and QUIN/KYNA (FC: 1.47, *p* = 0.16) tended to be higher in AKS subjects (Figure 2a).

No PLS-DA model was robust enough to discriminate between ALS and control subjects (Figure 2b). The scores plot showed that ALS and controls were separated with modest accuracy (62%) and the permutation test was not significant (*p* = 0.3). Lysine, KYN, KYNA, 3-HK/KYNA, and asparagine were the features with the highest VIP values (Figure 2c).

### 3.4. KP Intermediates Modestly Associated with ALS Characteristics

Based on univariate analysis, no metabolite concentration differed significantly between subgroups of ALS patients and no PLS-DA model was significant according to the permutation test. The models did, however, enable the identification of putative relevant candidates.

Regarding the site of onset, arginine (FC:0.81, *p* = 0.07), tryptophan (FC:0.80, *p* = 0.10), methionine (FC:0.82, *p* = 0.12), and valine (FC:0.83, *p* = 0.13) concentrations appeared to be lower in the bulbar form (the more severe form), while 3-HK/KYN (FC: 1.44, *p* = 0.13), 3-HK/KYNA (FC: 1.44, *p* = 0.19), and QUIN/KYN ratios (FC: 1.35, *p* = 0.19) tended to be higher (Appendix A). The scores plot of the PLS-DA model discriminated the different sites at onset with good accuracy (80%), but the model was not significant (*p*-value of the permutation test = 0.2) (Appendix A). The most important features for the model were the QUIN/KYN, QUIN/KYNA, and 3-HK/KYN ratios, indole-3-acetic acid, and 3-HK (Appendix A).

Five metabolites were highlighted in the volcano plot of younger to older patients (Appendix A). The PLS-DA model was not sufficiently robust to discriminate the two groups, and it had modest accuracy (63%) and the permutation test was not significant (*p* = 0.2) (Appendix A). 5-OH tryptophan, QUIN, glutamic acid, 5-HIAA, and arginine, however, were the most important variables in this modest discrimination (Appendix A).

Concerning the loss of weight at diagnosis, 2 metabolites were displayed in the volcano plot (Appendix A). The PLS-DA model was not robust (accuracy = 44%, *p*-value of the permutation test = 0.4) (Appendix A) and the features with the biggest VIP values were arginine, histidine, glutamic acid, QUIN, and asparagine (Appendix A).

As for ASLFRS-r score at diagnosis, no features were highlighted in the volcano plot with a FC threshold of 1.2 and raw *p*-value threshold of 0.2 (Appendix A). However, this can be explained by the weak variation of ALSFRS-r (median = 40) score between the groups. The two groups were not accurately separated on the score plot of the PLS-DA model (accuracy = 65%, *p*-value of the permutation test = 1) (Appendix A). The most important metabolites were threonine, phenylalanine, methionine, valine, and ornithine (Appendix A).

Concerning FVC, no features were highlighted in the volcano plot with a FC threshold of 1.2 and raw *p*-value threshold of 0.2 which can also be explained by the weak variation (median FVC = 96%) between the groups (Appendix A). With PLSDA, they were discriminated but with modest accuracy (48%) (permutation test: *p* = 0.9) (Appendix A) and the most valuable features were 3-HK, glycine, KYN, QUIN, and 5-HIAA. (Appendix A).

The Venn diagram (including metabolites with VIP > 0.8 in the different models exploring age at onset of symptoms, weight loss, FVC, and ALSFRS-r score at diagnosis) is shown in Appendix A.

### 3.5. Some Promising Prognosis Biomarker among Amino Acids or KP Metabolites

Univariate analyses showed neither amino acids nor kynurenine pathway intermediates as significantly different after *p*-value adjustment between the groups according to parameters chosen to represent disease evolution, except for 5-OH tryptophan being higher in patients who lost more weight at 12 months. No robust PLS-DA model was obtained, and permutation tests were not significant, but the modest discriminations enabled the identification of relevant compounds associated with disease evolution.

#### 3.5.1. Weight Loss at 12 Months (n = 21)

Briefly, 6 features were displayed on the volcano plot between the two groups according to the median (1.49% loss) (Appendix A). Score plots of PLS-DA discriminated between the two groups with a correct accuracy of 76%, but the permutation test was not significant (*p* = 0.2) (Appendix A). 5-OH tryptophan, indole 3-acetic acid, glycine, QUIN and alanine were the most important metabolites in the discrimination (Appendix A).

#### 3.5.2. ALSFRS-r at 12 Months (n = 22)

Five metabolites were highlighted on the volcano plot between the two groups according to the median (32% of ALSFRS-r score reduction) (Appendix A). Good separation was observed on the score plot of the PLS-DA model, but accuracy was poor (64%) (permutation test: *p* = 0.7) (Appendix A). The most important features were 5-HIAA, glycine, QUIN, arginine, and indole-3-acetic acid (Appendix A).

#### 3.5.3. FVC at 12 Months (n = 9)

FVC at 12 months and CSF profiles were available for only 9 ALS patients because the remaining were unable to blow. Therefore, even though we analyzed this parameter, statistical power was too weak for the results to be mentioned (Appendix A).

#### 3.5.4. Duration of Disease (n = 25)

For duration of disease, CSF concentrations of arginine (FC: 0.82, *p* = 0.04) and serine (FC: 0.82, *p* = 0.09) seemed to be lower in patients who died early (within 22.6 months), according to the volcano plot (Figure 3a). Good separation was obtained between the two groups. The accuracy of the model was acceptable (79%), although the permutation test was not significant (*p* = 0.6) (Figure 3b). Arginine, 5-OH tryptophan, serine, aspartic acid, and lysine were the most discriminant (Figure 3c).

#### 3.5.5. Combined Parameters of Prognosis

A Venn diagram built for these 3 prognosis parameters and survival time is shown in Figure 3d. In general, only few features were specific to one parameter, and QUIN, 3-HK, 5-OH tryptophan, 5-HIAA, glycine, and arginine were common to all four parameters. Some metabolites were also common to 3 parameters: KYN, asparagine, indole-3-acetic acid, serotonin, lysine, and serine.

Features having a VIP value > 0.8 in either PLS-DA model between ALS and control subjects or for prognostic parameters at 12 months have been highlighted in the metabolic pathway of Figure 1.

We also conducted a multiparametric survival analysis of disease duration with the 12 metabolites commonly important to at least 3 parameters. Arginine, QUIN, 5-OH tryptophan, and 5-HIAA had a *p*-value < 0.2 with the univariate Cox model. None was significantly correlated with ALS duration in multivariate analysis after correction by the age of onset.

## 4. Discussion

### 4.1. Validation of Analytical Methods Used in Routine Practice for Research Projects

We chose to use a targeted approach to explore relevant pathways guided by the literature. The major strength of our study was the use of robust, validated, and routinely available analytical methods, which are accredited and even met the criteria of the International Organization for Standardization 15189 for amino acid determination. This is a key element since it allows for the immediate use in clinical practice of any biomarker discovered using this approach. To our knowledge, this is the first study to use a univariate followed by a multivariate analysis approach focused on CSF tryptophan metabolism intermediates in the context of ALS, in combination with a validated method of amino acid quantification.

### 4.2. No Benefit to Early Exploration of CSF Amino Acids in ALS

The value of CSF amino acids exploration in ALS is the subject of long-standing debate. Numerous studies have explored amino acid profiles in the CSF of ALS patients, with controversial results. In 1990, Rothstein et al. found higher concentrations of aspartic acid, threonine, serine, glutamic acid, and lysine in the CSF of early diagnosed ALS patients compared to control subjects [16], while Perry et al. found elevated glutamine, alanine, valine, leucine, tryptophan, lysine, and arginine levels [17]. Other studies conducted over the same period, however, failed to find any significant differences in concentrations at diagnosis [18,19]. The controversial results of these studies can be explained in part by the small number of ALS patients (always fewer than 20) and by the difference in statistical methods employed (i.e., corrections for multiple tests, rarely done in small cohorts). More recently, Valentino et al. found higher concentrations of homocysteine in a larger cohort of ALS subjects (69 ALS vs. 76 controls with other neurological disorders) in both plasma and CSF, but with no difference within each group of patients [20]. Interest in CSF amino acids in ALS have waned over the past decades, and metabolomics studies including amino acids explorations have been largely performed on ALS in different biological fluids. In 2010, we investigated the CSF metabolomic profile (including glutamine, alanine, and tyrosine) of 50 ALS patients by nuclear magnetic resonance (NMR) and did not find any significant differences of these 3 amino acids [21]. In a cohort of 22 ALS patients compared to 18 controls (patients with disorders other than neurodegenerative diseases) and 22 PD subjects (using an untargeted approach using the combination of NMR, liquid chromatography coupled with tandem mass spectrometry (LC-MS/MS) and gas chromatography coupled to tandem mass spectrometry (GC-MS/MS)) isoleucine, valine, and alanine appeared to be higher in ALS than in controls, without being as different as in PD subjects [22,23]. Importantly, no clear consistency has been observed between these different studies reporting AA explorations. This may be due, as usual, to population heterogeneity (both of ALS and controls), time of sample collection, presence of treatment, and differences in analytical performances.

Here we determined the CSF concentrations of 22 amino acids in 40 ALS patients and 42 control subjects, using a robust and validated liquid chromatography coupled with mass spectrometry with a derivatization method. In our cohort, amino acid concentrations were similar between cases and controls in both univariate and multivariate analyses. Contrary to many studies previously published, we used the correction for multiple tests and based the reliability of the multivariate models on a permutation test. Multivariate analysis, however, remained relevant to highlight some metabolites. Lysine, asparagine, 4-OH proline, phenylalanine, glutamine, arginine, tryptophan, aspartic acid, serine, alanine, and taurine had a VIP value > 0.8 within the first component of the PLS-DA model between cases and subjects. To note, increased concentrations of glutamic acid were found in about 25% of patients. This heterogeneity was observed in both ALS and control subjects and could not be explained by the analytical incertitude of our method [24]. Many investigations assessing glutamic acid levels in the CSF of ALS subjects have been performed but its relationship with glutamate-mediated excitotoxicity is unclear [25]. Considering the heterogeneity of both ALS patients and controls, all suffering from brain disorders, we suggest that glutamic acid, i.e., one of the most important neurotransmitters is highly variable. This observation has been previously reported in a subset of patients from motoneuron disease, including ALS [26]. However, similarly to this study, the heterogeneity remained unexplained in our cohort.

As revealed by Figure 1, most amino acids tended to be decreased in patients who had the worst evolution. One of the major benefits of this study was the exploration of patients at the time of diagnosis, before initiation of any etiological and symptomatic treatment. Changes in amino acid concentrations, however, have been observed throughout ALS disease. An increase in asparagine, isoleucine, leucine, phenylalanine, and tryptophan, and a decrease in valine were observed between two CSF samples 12 months apart among the same ALS patients [18]. These findings suggest that our negative findings might be explained by the early time of sample collection. To avoid the performance multiple lumbar punctures, amino acids profile follow-ups may be planned in plasma, as part of the patient’s routine biological explorations base on an accredited method.

Altogether, amino acids in CSF do not seem to be useful as diagnostic or prognostic biomarkers of ALS at disease onset. However, their diagnosis potential in plasma has recently been demonstrated in a study using machine learning approach where methylhistidine was found to be significantly lower in ALS patients (n = 134) and seven essential amino acids were also lightly decreased in male patients, although their correlation with ALSFRS score was poor [27].

### 4.3. Contradictory Findings on Kynurenine Pathway

We also investigated intermediates of tryptophan metabolism into NAD+ (via the KP) and melatonin. To the best of our knowledge, very few studies have previously explored the KP. In 2003, Iłzecka et al. measured levels of kynurenic acid in 16 CSF samples (at diagnosis) and 14 plasma samples of ALS and compared its concentration to patients with acute vasomotor headache, with no differences between both populations. Kynurenic acid levels, however, were higher in the CSF of severe forms and bulbar onset than in controls, and lower in serum. The constitution of patient subsets in a cohort of 14, however, clearly reduced the power of this analysis [11]. The most significant results were obtained by Chen et al. who explored serum and CSF KP [10] in a larger cohort of 140 ALS patients vs. 35 controls (with suspected meningitis but without any disease found during follow-up examinations). They mainly observed a two-fold increase in tryptophan concentration in the serum and CSF of ALS patients, and two- and ten-fold increases in kynurenine in serum and CSF, respectively, along with a four-fold increase in IDO-1 activity in CSF. Another recent study failed to find any significant differences [12]. As already discussed by Janssens et al., the significant results in the work of Chen et al. could be attributable to their use of non-age-matched control and ALS groups: Kynurenine pathway activation increases with ageing [12]. Age could therefore explain the lack of difference found by the more recent studies, such as ours, which used age-matched control and ALS groups. Consistently, no compound of the kynurenine pathway was significantly altered in our cohort between ALS patients and control subjects, despite a tendency for KYN, KYNA, and QUINA to be lower in ALS patients, while 3-HK/KYNA and QUIN/KYNA ratios tended to be higher according to the volcano plot. As QUIN and 3-HK may be neurotoxic and KYNA neuroprotective, these findings could reflect a KP impairment toward neurotoxic compounds, but this analysis is not powerful enough. Moreover, the investigation period in the study by Chen et al. was different: patients were already treated by riluzole, contrary to our study where CSF was collected at the time of diagnosis in patients naive to treatment. This raises doubts concerning the origin of KP modification, which may be induced by the anti-glutamatergic effect of riluzole or by disease evolution. Moreover, for ethical reasons, our control subjects had other neurological diseases from which neuroinflammation and alterations of KP cannot be excluded. These other neurological diseases included neuropathy without etiology or iatrogenic neuropathy, chronic inflammatory demyelinating polyradiculoneuropathies, or neuropathies associated with dysglobulinemia. As expected for a study based on CSF exploration, our study did not include healthy controls. However, most investigations cited in this discussion have been conducted in the same way (with the exception of the Chen et al. study [10]). The comparison of ALS patients to patients with other neurological disorders makes sense in terms of evaluating the ability to detect more specific biomarkers. Thus, it appears that early exploration of KP in CSF does not differentiate ALS from other neurological diseases.

While neuroinflammation is uniformly present in end-stage pathology and is recognized as a major player in ALS progression [28], how and when it occurs remains unclear due to a lack of studies investigating this phenomenon in early disease and over the duration of the disease [29]. In fact, it was recently suggested that initial pre-symptomatic and early symptomatic phases are dominated by anti-inflammatory immune responses, whereas late symptomatic and terminal stages are dominated by proinflammatory immune responses [30]. This theory could actually explain the absence of KP modification found at early stages (diagnosis) in ALS patients, while most intermediates tend to be higher in ALS subjects with more severe evolution. To note, 3-HK, as well as 3-HK/KYNA and QUIN/KYNA ratios (neurotoxic indexes) also seemed higher in bulbar forms, known to be more severe than spinal forms. The QUIN/KYNA ratio tended to be higher in ALS subjects who died within 22.6 months, and this result was independent from the age of onset (correlation between and QUIN/KYNA ratio: r^2^ = −0.028, *p* = 0.86). However, this ratio was not relevant for the other parameters of disease progression. As some parameters of evolution were not correctly taken into account in this study (low loss of weight over 12 months, missing FVC values), the survival time is clearly the most important criteria.

Apart from KP and serotonin, TRP is also catabolized by another metabolic pathway: Direct metabolism into indole and derivates by the gut microbiota [31]. Indole metabolites, like indole-3-acetic acid, also participate in immune regulation [8]. To the best of our knowledge, no previous study has investigated the relationship between ALS and these indole intermediates in CSF. In our cohort, none were significantly different. In recent years, the microbiota has been the subject of much attention, suggesting its link to a vast range of diseases, including neurodegenerative disorders [32,33]. Moreover, the intestinal microbiota may regulate circulating tryptophan availability and kynurenine pathway metabolism in the periphery and the brain [33,34]. Concerning ALS, as recently reviewed by Erber et al., very few studies have been conducted in humans, covering a total of 145 patients with ALS [35]. Although the gut microbiota appear to be a key mediator of the brain–gut axis and play a role in many aspects of brain function, no consistent result has been found yet in humans. Considering interactions between theses pathways, the concomitant exploration of tryptophan pathway and microbiota should be planned during the evolution of ALS patients.

### 4.4. Major Candidates among AA and KP Metabolites

Interestingly, the Venn diagram revealed that major KP intermediates (KYN, 3-HK, QUIN) were important variables for at least three disease evolution parameters, and the others, as well as most metabolite ratios, were relevant for at least one disease evolution parameter. Without being significant, these intermediates tended to be higher in patients displaying the highest variation in these parameters, sometimes with important FC. These findings are consistent with the results of Iłzecka et al. in the severe forms. Moreover, the trend of the 3-HK/KYNA and QUIN/KYNA ratio to be higher in ALS patients and in bulbar forms may reflect KP impairment toward production of 3-HK and QUIN, which are neurotoxic compounds, in detriment of KYNA, which is neuroprotective. The lack of significance of the permutation tests may be explained by the lack of statistical power for each model, which is largely compensated by the observation of common candidates in these models. These findings are promising and merit to be included in a larger cohort to increase the power of the results and to determine the role of these candidates in disease evolution.

## 5. Conclusions

To conclude, we did not highlight any early modifications of amino acids or the kynurenine pathway in ALS patients in our population. These findings are not in favor to the use of CSF amino acid concentrations and tryptophan metabolism intermediates for early diagnosis of ALS patients. However, we observed an elevated (without being significant) neurotoxicity index in ALS patients and in bulbar forms, as well as a general trend of KP intermediates and QUIN /KYNA ratio being higher in patients who developed severe forms. Moreover, it underlined the importance of exploring in more depth the kinetics of inflammation associated to KP in ALS. More generally, we suggest not systematically focusing the biological exploration on disease onset but to monitor these parameters, in order to establish some important steps of evolution, useful to suggest targeted personalized therapeutics. Importantly, this work provides the opportunity to explore microbiota also highly linked with KP and inflammation which has been of particular interest for researchers in ALS in the past few years.

## Figures and Tables

**Figure 1 biomolecules-11-00691-f001:**
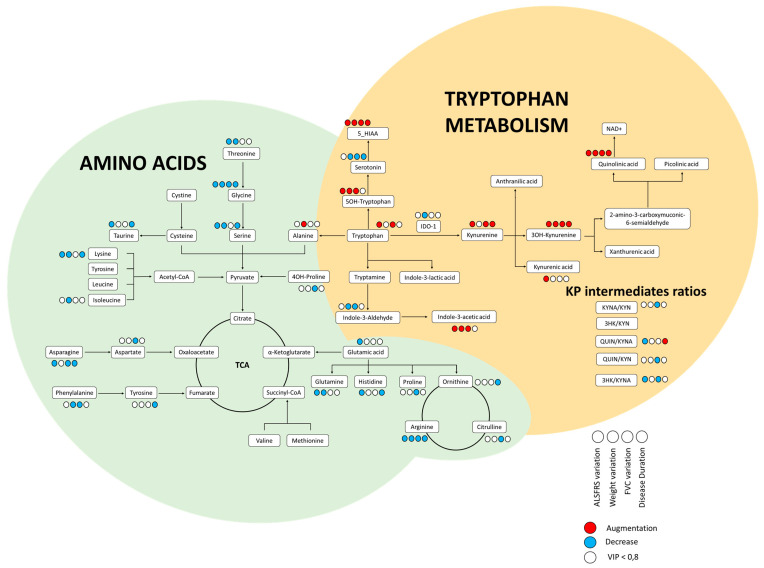
Metabolic pathway explored in this work. Colored dots are present only when the metabolite had a VIP value > 0.8 for one of the following parameters: ALSFRS variation (first dot), weight loss (second dot), FVC over year (third dot), survival (fourth doth). A blue dot indicates a decrease of the feature and a red dot an increase. TCA: Tricarboxylic acids.

**Figure 2 biomolecules-11-00691-f002:**
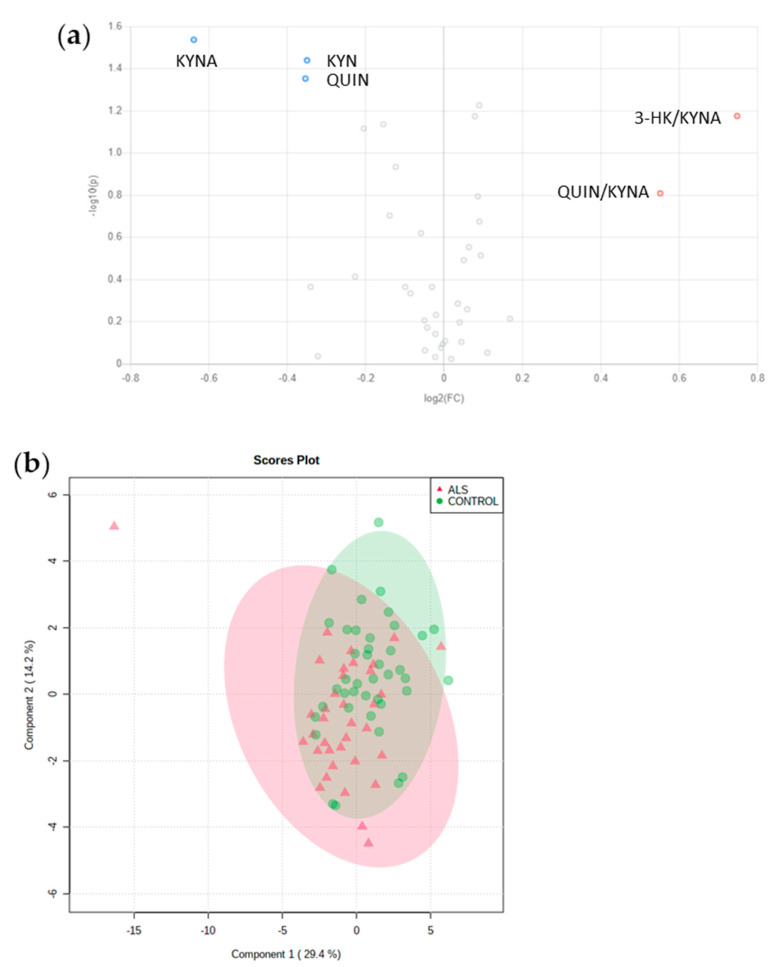
Multivariate analysis of CSF amino acids and tryptophan catabolism metabolites in ALS and control subjects. (**a**) Volcano plot representing the most important features in univariate analysis. Red dots indicate an increase of the feature in ALS patients compared to controls and blue dots a decrease; (**b**) scores plots of the PLS-DA model. Red triangles represent ALS patients and green dots control subjects. (**c**) Important features (VIP > 0.8) identified by PLS-DA, the boxes on the right indicate the relative concentrations of the corresponding metabolite in each group.

**Figure 3 biomolecules-11-00691-f003:**
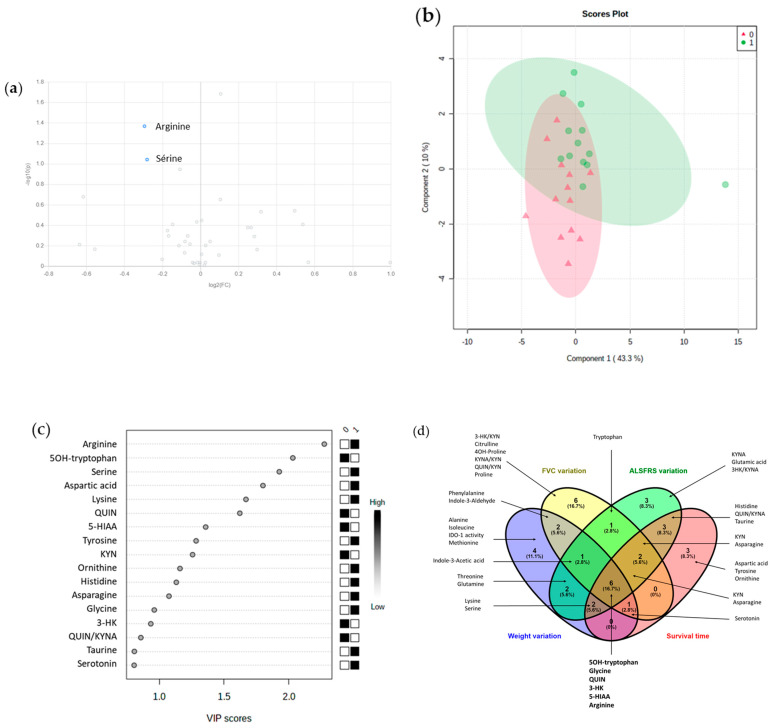
Multivariate analysis of CSF amino acids and tryptophan catabolism metabolites for prognostic biomarkers of ALS. Only figures showing results for the duration of disease are represented on panels (**a**), (**b**), and (**c**). Two groups were obtained according to the median duration until death (22.6 months). (**a**) Volcano plot representing the most important features in univariate analysis. Blue dots indicated a decrease of the feature in patients who died within 22.6 months after the first symptoms; (**b**) scores plots of the PLS-DA model. Red triangles (0) represent ASL patients who died within 22.6 months after the first symptoms and green dots (1) represents patients who died after 22.6 months; (**c**) important features (VIP > 0.8) identified by PLS-DA. The boxes on the right indicate the relative concentrations of the corresponding metabolite in each group. (**d**) Venn diagram representing metabolites with a VIP > 0.8 with PLS-DA models between ALS patients according to variation of weight, FVC and ALSFRS-r scores over a year, and survival time.

**Table 1 biomolecules-11-00691-t001:** Characteristics of ALS patients and controls.

	ALS Patients	Control Subjects	*p*-Value
N	40	42	
Gender (% female)	42.5	28.6	0.28
Age at sample collection (mean ± SD)	65.3 ± 12.4	68.0 ± 12.43	0.33
Age at onset (mean ± SD)	64.4 ± 12.5		
BMI (kg/m2) (mean ± SD)			
At diagnosis	23.2 ± 4.2		
At 12 months	22.6 ± 5.2		
Weight loss at diagnosis (%) (mean ± SD)	4.54 ± 6.5		
Diagnostic delay (month) (mean ± SD)	9.0 ± 5.8		
Site at onset (%)			
bulbar	27.5		
spinal	72.5		
ALSFRS-r score (mean ± SD)			
At diagnosis	39.8 ± 5.3		
At 12 months	28.2 ± 9.8		
FVC (%) (mean ± SD)			
At diagnosis	94.5 ± 26.4		
At 12 months	85.2 ± 24.7		
Disease duration (months) (median, quartiles)	25.0 (18.1–38.8)		

Abbreviations: N: Numbers; ALS: Amyotrophic lateral sclerosis; SD: Standard deviation; BMI: Body mass index; ALSFRS-r: ALS Functional Rating Scale—Revised, FVC: Forced vital capacity.

**Table 2 biomolecules-11-00691-t002:** Compounds found in the CSF of ALS and control subjects, before normalization.

	**ALS Patients**	**Control Subjects**	**Raw *p*-Value**
**N**	**40**	**42**	
	**Amino Acids (Mean ± SD) (µM)**		
Aspartic acid	1.4 ± 1.0	1.1 ± 0.2	0.88
Asparagine	7.3 ± 2.1	7.0 ± 1.8	0.46
4-OH proline	3.3 ± 0.6	3.0 ± 0.7	0.20
Proline	1.2 ± 0.4	1.5 ± 1.3	0.62
Ornithine	3.5 ± 1.8	3.4 ± 1.4	0.79
Threonine	29.9 ± 9.2	30.0 ± 8.5	0.97
Glycine	5.7 ± 2.9	5.4 ± 2.8	0.48
Citrulline	4.4 ± 1.1	4.4 ± 1.3	0.74
Glutamic acid	26.8 ± 48.9	22.9 ± 45.9	0.61
Alanine	41.3 ± 13.0	41.1 ± 12.8	0.88
Cystine	<1	<1	1
Methionine	3.2 ± 1.1	3.2 ± 1.0	0.75
Tyrosine	9.4 ± 3.3	9.8 ± 3.0	0.53
Histidine	10.8 ± 3.4	10.9 ± 3.1	0.44
Lysine	28.3 ± 6.6	27.1 ± 7.6	0.29
Arginine	17.2 ± 4.9	16.5 ± 5.0	0.14
Leucine	15.8 ± 5.6	16.3 ± 4.5	0.37
Isoleucine	5.8 ± 2.5	5.9 ± 2.2	0.56
Taurine	8.3 ± 1.9	8.2 ± 1.8	0.91
Serine	20.5 ± 5.6	19.3 ± 5.0	0.40
Homocysteine	<1	<1	1
Phenylalanine	10.7 ± 2.9	10.4 ± 2.4	0.52
Glutamine	523.3 ± 96.3	515.7 ± 89.8	0.75
Valine	20.2 ± 7.2	21.7 ± 7.1	0.34
	**ALS Patients**	**Control Subjects**	**Raw *p*-Value**
**N**	**36**	**41**	
	**Tryptophan Metabolism Intermediates (Mean ± SD) (nM)**		
3-HK	35.0 ± 14.0	40.9 ± 24.1	0.380
QUIN	27.1 ± 18.9	34.6 ± 20.1	0.044
Serotonin	69.9 ± 9.8	67.5 ± 7.7	0.32
5-OH tryptophan	7.4 ± 2.7	7.1 ± 2.1	0.55
KYN	44.5 ± 24.0	56.7 ± 26.1	0.036
Tryptophan	1671.2 ± 639.2	1840.2 ± 586.9	0.20
5-HIAA	96.9 ± 63.4	100.4 ± 52.7	0.62
KYNA	1.1 ± 1.4	1.7 ± 1.7	0.029
Indole 3 lactic acid	5.5 ± 0.04	6.0 ± 1.8	0.11
Indole 3 aldehyde	2.4 ± 0.3	2.4 ± 0.2	0.72
Indole 3 acetic acid	28.6 ± 46.2	31.9 ± 22.9	0.072
IDO-1 activity	0.027 ± 0.01	0.031 ± 0.01	0.076
KYNA/KYN	0.029 ± 0.038	0.030 ± 0.026	0.24
3-HK/KYN	1.0 ± 0.64	0.95 ± 1.17	0.059
QUIN/KYNA	54.5 ± 47.3	37.2 ± 32.2	0.15
QUIN/KYN	0.68 ± 0.33	0.63 ± 0.28	0.88
3-HK/KYNA	84.5 ± 79	50.3 ± 47.8	0.067

Abbreviations: ALS: Amyotrophic lateral sclerosis, 5-HIAA: 5-OH indole acetic acid, IDO-1: Indoleamine 2,3 dioxygenase 1, KYNA: Kynurenic acid, KYN: Kynurenine, 3-HK: 3-OH kynurenine, QUIN: Quinolinic acid.

## Data Availability

The datasets generated during and/or analyzed during the current study are available from the corresponding author on reasonable request.

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
