# Peer review of "Some CSF Kynurenine Pathway Intermediates Associated with Disease Evolution in Amyotrophic Lateral Sclerosis"

_biomolecules, 2021, doi:10.3390/biom11050691_

Round 1

Reviewer 1 Report

This is a well conducted negative study which evaluates Kynurenine Pathway components and amino-acids within CSF from ALS patients and age/sex matched disease controls. Whilst the headline results are negative the high standard of methodology and analysis makes this useful data for the field.  The weaknesses are lack of neurological normal controls and the relatively small sample size.

Minor edits:

  • Would benefit from inclusion of neurologically normal controls to help evaluate the biological significance of ALS-associated changes;  although I accept that disease controls are more relevant for design of a diagnostic biomarker.  As a minimum the absence of normal controls should be mentioned in the Discussion; for example this impacts comparison with previous literature.
  • "This section may be divided by subheadings. It should provide a concise and precise description of the experimental results, their interpretation, as well as the experimental conclusions that can be drawn." should be deleted from the Results -- I suspect this was missed from proofing.
  • Please remove the sentence "it is unethical to perform multiple lumbar punctures"; the ethical acceptance is study and participant dependent and multiple studies exist including serial lumbar punctures.

Author Response

Numbers of page and lines are relevant when all Track changes are displayed.

Minor edits :

  • Would benefit from inclusion of neurologically normal controls to help evaluate the biological significance of ALS-associated changes; although I accept that disease controls are more relevant for design of a diagnostic biomarker.  As a minimum the absence of normal controls should be mentioned in the Discussion; for example this impacts comparison with previous literature.

We thank the reviewer for his relevant comment. We could not include healthy controls in our study for ethical reasons. We added a paragraph in the section 4.3 to mention the absence of normal controls and to remind the interest of disease controls in biomarker discovery (see p.14 l.477-483). Most of the studies involving CSF exploration are based on patients suffering from neurological diseases. However, it seems more relevant to compare diseases with high difficulty to establish a diagnosis, with mimic diseases or diseases affecting the same tissue. We specified the clinical characteristics of the control subjects included in the previous investigations discussed in the paper (see section 4.2, p.12 l.393; section 4.2, p.12 l.400-401; section 4.3, p.13 l.450-451; section 4.3, p.13 l.456-457).

  • "This section may be divided by subheadings. It should provide a concise and precise description of the experimental results, their interpretation, as well as the experimental conclusions that can be drawn." should be deleted from the Results -- I suspect this was missed from proofing.

We thank the reviewer for his comment and made the correction (see p.4, l.194-196).

  • Please remove the sentence "it is unethical to perform multiple lumbar punctures"; the ethical acceptance is study and participant dependent and multiple studies exist including serial lumbar punctures.

We thank the reviewer for his relevant comment. This sentence has been replaced by “To avoid  the  performance multiple lumbar punctures…” (see section 4.2 p.13, l.435-436).

Reviewer 2 Report

This is a complex and interesting study. The design appears to be appropriate and the results are clearly presented. I have only minor comments:

  1. On some occasions the authors mention trends while actually certain results are actually non-significant. Perhaps the concept of trend could be used in a less permissive manner (i.e. on fewer occasions).
  2. One could note that the CSF levels of glutamic acid are very variable, both in ALS patients and in controls. Perhaps this could need to be justified and commented on.

Author Response

Numbers of page and lines are relevant when all Track changes are displayed.

Minor edits :

  • On some occasions the authors mention trends while actually certain results are actually non-significant. Perhaps the concept of trend could be used in a less permissive manner (i.e. on fewer occasions).

We thank the reviewer for his relevant comment. We restrained the concept of the trend for the more important comparisons (i.e. ALS patients vs control, the onset site and the survival time) and removed it from the others parameters (see section 3.4, p.9, l.271-275; section 3.4, p.9, l.280-282; section 3.5.1, p.9-10 l.312-316; section 3.5.2 p.10 l.322-326; section 4.3 p.14 l.506-507 ). The use of p-value adjustment for multiple comparison is restrictive and limit the opportunity to identify significant biomarkers identification in a complex disease like ALS, especially in a modest cohort like ours. We believe that the description of tendencies may be interesting to discuss the findings of other groups who did not apply correction for multiple tests. However, we must be aware of the limit of such discussion and we must limit the scope of the results.

  • One could note that the CSF levels of glutamic acid are very variable, both in ALS patients and in controls. Perhaps this could need to be justified and commented on

We thank the reviewer for his relevant comment. A paragraph has been added in section 4.2 to comment this heterogeneity (see p.12-13, l.418-427). We recognize that, like others before us, we do not have the clear explanation for this inter-individual variability of CSF glutamic acid within each group of subjects.